# Determinants of Changes in Women’s and Men’s Physical Activity and Sedentary Behavior across the Transition to Parenthood: A Focus Group Study

**DOI:** 10.3390/ijerph19042421

**Published:** 2022-02-19

**Authors:** Vickà Versele, Femke Marijn Stok, Anna Dieberger, Tom Deliens, Dirk Aerenhouts, Benedicte Deforche, Annick Bogaerts, Roland Devlieger, Peter Clarys

**Affiliations:** 1Department of Movement and Sport Sciences, Faculty of Physical Education and Physiotherapy, Vrije Universiteit Brussel, Pleinlaan 2, 1050 Brussels, Belgium; tom.deliens@vub.be (T.D.); dirk.aerenhouts@vub.be (D.A.); benedicte.deforche@ugent.be (B.D.); peter.clarys@vub.be (P.C.); 2Department of Development and Regeneration, Faculty of Medicine, KU Leuven, Herestraat 49, 3000 Leuven, Belgium; annick.bogaerts@kuleuven.be (A.B.); roland.devlieger@uzleuven.be (R.D.); 3Department of Interdisciplinary Social Science, Utrecht University, Heidelberglaan 1, 3584 CS Utrecht, The Netherlands; f.m.stok@uu.nl; 4Department of Obstetrics and Gynaecology, Medical University of Graz, Auenbruggerplatz 14, 8036 Graz, Austria; anna.dieberger@medunigraz.at; 5Department of Public Health and Primary Care, Faculty of Medicine and Health Science, Ghent University, C. Heymanslaan 10, 9000 Ghent, Belgium; 6Centre for Research and Innovation in Care (CRIC), Faculty of Medicine and Health Sciences, University of Antwerp, Universiteitsplein 1, 2610 Wilrijk, Belgium; 7Faculty of Health, University of Plymouth, Plymouth PL4 8AA, UK; 8Obstetrics and Gynaecology, University Hospitals, KU Leuven, Herestraat 49, 3000 Leuven, Belgium; 9Department of Obstetrics, Gynaecology and Fertility, GZA Campus Wilrijk, Oosterveldlaan 24, 2610 Wilrijk, Belgium

**Keywords:** pregnancy, postpartum, active lifestyle, qualitative research, behavior change

## Abstract

Background: Becoming a parent may cohere with drastic changes in physical activity (PA) and sedentary behavior (SB). A clear understanding of determinants of changes in PA and SB during pregnancy and postpartum is needed to facilitate the development of tailored family-based interventions. Methods: Thirteen focus group discussions targeting determinants of changes in PA and SB behavior were conducted, involving a total of 74 expecting and first-time parents. A semi-structured question guide was used to facilitate the discussions. Results: Four main levels of determinants were identified: the individual (including psychological, situational and biological determinants), interpersonal, environmental and policy level. Some determinants were mentioned to be a barrier (e.g., “barriers to self-care”) while others were a facilitator (e.g., “weight control”). Determinants were related to both PA and SB and applicable during pregnancy as well as postpartum (e.g., “self-regulation”), or only related to one behavior and/or one period (e.g., “feeding baby”). Some were described by both parents (e.g., “parenthood perceptions”), whereas others were mentioned by women (e.g., “PA knowledge”) or men (e.g., “time opportunities”) only. Conclusions: Focus should be given to interventions aimed at improving parents’ self-regulation skills and support on how to cope with interpersonal and situational constraints as well as parenthood perceptions.

## 1. Introduction

Engaging in sufficient levels of physical activity (PA) and limiting sedentary behavior (SB) during and after pregnancy is essential to promote the health of both mother and child. Sufficient PA during pregnancy is related to a reduction in the risk of developing gestational diabetes mellitus, hypertensive disorders and depressive symptoms, and can help reduce excessive weight gain during pregnancy as well as prevent weight retention in the postpartum period [1,2,3,4]. It has also been associated with a reduction in cesarean section rates [5] and reduced odds of macrosomia, without an increase in preterm birth rates or low-birthweight offspring [6]. While SB during pregnancy is less well studied, it has been shown that more sedentary time during pregnancy is associated with an abnormal maternal glucose metabolism [7,8], increased neonatal adiposity [9] and fetal macrosomia [10].

Based on this evidence, adults are advised to participate in at least 150 min of moderate-intensity PA per week and to limit their sedentary time [11]. Although it has been shown that pregnant women perceive PA during pregnancy as important, beneficial and safe [12,13], many do not reach the recommended PA levels [14,15]. Indeed, most pregnant women decrease their PA levels and increase their SB levels over the course of their pregnancy [16], while PA levels remain low postpartum [16,17,18,19]. As for fathers-to-be, the pregnancy period has also been described as a stressful period during which they might experience similar mental, physical and lifestyle changes [20,21,22,23]. As a consequence, maintaining PA levels and limiting SB is equally important for them. Furthermore, partners are an important source of support for lifestyle behaviors of pregnant women [24].

Addressing unfavorable changes in PA and SB across the transition to parenthood is of high importance, not only due to the associated short-term adverse health outcomes (e.g., increased risk for gestational diabetes, cesarean section, macrosomia) [1,2,3,4,5,6]. If desired PA levels are not restored in the postpartum period, the lack of PA may persist during the rest of the life span [25,26]. As future role models for their children, parents’ behavior will directly influence the health status of the next generation, and thus encouraging parents to adopt healthy behaviors should be prioritized. In addition, during pregnancy and the immediate postpartum period parents are in frequent contact with healthcare providers, which gives an opportunity for professional advice towards health behavior. Moreover, people experiencing life-changing events could be an important target group for lifestyle interventions, which makes this transition period a window of opportunity for behavior change [18,27].

In summary, as both maternal and paternal PA and SB are important during and after pregnancy, a full understanding of the barriers and facilitators of changes in PA and SB across the transition to parenthood is needed. This is important to prevent undesirable changes during this transition period. Perceived barriers to PA have been described and include intrapersonal (e.g., pregnancy-related symptoms and limitations), interpersonal (e.g., lack of advice and information) and environmental (e.g., adverse weather) determinants [12,28]. However, most barriers are often described in relation to PA and to pregnant women only. SB has not been studied widely, and the impact of (upcoming) fatherhood on men’s lifestyle behaviors has also largely been neglected. Furthermore, most research has focused on the behavior itself, with no research available on behavior change. While many studies have assessed PA during and after pregnancy, studies about determinants of changes in PA and SB behavior in both women and men during and after pregnancy are needed to move towards effective interventions aimed at increasing (expecting) parents’ PA while limiting their SB levels. Therefore, the objectives of this study were to identify determinants of changes in PA and SB across the transition to parenthood (i.e., during pregnancy up to one year postpartum) in both women and men.

## 2. Materials and Methods

A qualitative research design using focus group discussions was used for data collection. The questionnaire was developed, pre-tested and validated by experts (face validity) with experience in focus-group methodology and health behavior. Participants were recruited through a snowball sampling strategy [29]. The research described in this article has focused on couples consisting of a mother and a father because of differences in energy metabolism and PA behaviors between women and men [30,31]. The exclusion of same-sex couples allows a homogeneous comparable “partner group”. Therefore, the following terminology will be used to refer to the participants: “(pregnant) women”/“mothers”/and “men”/“fathers(-to-be)”. Two sets of focus groups were organized. For the first set, focusing on changes during pregnancy, women and men expecting their first child or with a first child less than three months old were recruited. For the second set, focusing on changes in the postpartum period, mothers and fathers with a first child of three months to one year old were recruited. For both sets of focus groups, participants were recruited individually or as a couple. Both homogeneous (same-sex) and heterogeneous (mixed-sex) focus group discussions were conducted to ensure diversity of both interactions and opinions within and across focus groups. All focus groups were conducted between March and June 2019. Through a preliminary content analysis after each focus group discussion, the point of theoretical saturation was determined, after which one extra focus group was organized to ensure true saturation. The final data analysis was performed after all focus groups were completed [32]. More details describing the design of this study have been described elsewhere [33]. The protocol of the study and related documents were approved by the Medical Ethics Committee of the University Hospital (Vrije Universiteit Brussel, Brussels, Belgium). The trial was conducted in compliance with the principles of the Declaration of Helsinki (current version), the principles of Good Clinical Practice and in accordance with all applicable regulatory requirements [34].

### 2.1. Question Guide

At the start of each focus-group discussion, participants signed an informed consent form and completed a short sociodemographic and lifestyle questionnaire. This included general questions on demographics (i.e., age and educational level: no degree, lower, secondary, college and university education), perceived health (“How would you currently describe your health status in general?” on a 5-point scale), perceived diet quality (“To what extent do you feel you are eating healthy?” on a 5-point scale), PA level (“How many days during the last week were you physically active for at least 30 min?”, 0–7 days), body weight (self-reported, in kg) and height (self-reported, in cm). All focus group discussions were conducted following a semi-structured question guide (see Table 1). Being part of a larger study, the question guide not only focused on PA and SB, but also on eating behavior [33]. For the current paper, we only report the questions and results on changes in PA and SB and their associated determinants. Participants were asked to answer the questions for changes in their own behavior and not for changes in their partners’ behavior.

### 2.2. Data Analysis

All focus group discussions were audio taped and transcribed verbatim in Microsoft Word using Windows Media Player 12. NVivo 12 software was used for qualitative data analysis. The same inductive thematic approach and procedure to derive sub- and main categories from the data was used as described in the twin paper on determinants of changes in eating behavior [33]. Quotes derived from question 1 were used to provide a framework of changes in PA/SB, and quotes derived from questions 2–3 were taken together as quotes reflecting determinants causing changes in PA/SB. All quotes were coded and grouped in (sub)units, after which they were refined and renamed into a list of determinants. This was completed independently by two researchers (VV and PD), and through several discussion rounds with experts in the field of PA and SB (VV, MS and TD) to ensure reliability of data synthesis. Although some overlap between categories of determinants might exist, clear cut-off choices were made in order to improve readability and clarity purposes. Based on this categorization, four frameworks were developed; two for PA (during pregnancy and postpartum), and two for SB (during pregnancy and postpartum). The full list of determinants of both behaviors and periods and their categorizations can be found in Appendix A. IBM SPSS Statistics 27.0 was used to calculate means and standard deviations from the quantitative data obtained from the questionnaires and to calculate descriptive statistics of the focus group sample.

## 3. Results

A total of 74 participants (48.6% males) participated in thirteen focus group discussions, of which seven focused on changes in PA and SB during pregnancy and six focused on PA and SB postpartum. All participants were Caucasian, mostly highly educated and physically healthy. Sample characteristics can be found in Table 2.

### 3.1. Changes in PA and SB during Pregnancy and Postpartum

Most participants described one or more changes in their PA and SB during either the entire or only during the pregnancy or postpartum period. An overview of the described changes is summarized in Table 3.

### 3.2. Determinants of Changes in PA and SB during Pregnancy and Postpartum

Similar to the twin paper on changes in eating behavior during the transition to parenthood [33] four frameworks were developed: (1) determinants of changes in PA during pregnancy (Figure 1) and (2) during the postpartum period (Figure 2); (3) determinants of changes in SB during pregnancy (Figure 3) and (4) during the postpartum period (Figure 4). The frameworks of PA consist of four main socioecological levels, i.e., the individual (subdivided into psychological, situational and biological), interpersonal (social), environmental (meso/macro) and the policy (governmental) level. The frameworks of SB consist of three main socioecological levels, i.e., the individual, interpersonal and policy level—no determinants were observed on the environmental level. For each determinant described below, it is indicated for which behavior it was described (PA and/or SB), and whether it was mentioned during pregnancy and/or postpartum, after which each (sub)category is illustrated by the most appropriate quotes (Table 4, Table 5, Table 6, Table 7, Table 8 and Table 9). Quotes from people who participated in the first set of focus groups were identified as ‘pregnant woman x’ or ‘expecting father x’, even though some already delivered at the time of the focus group discussion. Quotes from people who participated in the second set of focus groups were identified as ‘first-time mother x’ or ‘first-time father x’.

#### 3.2.1. Individual Level (PA/SB)

##### Psychological (PA/SB)

Physical activity knowledge—during pregnancy and postpartum (PA)

A lack of knowledge about which activities could be performed safely during pregnancy was mentioned. Pregnant women asked healthcare professionals for recommendations, explanations and reassurance (quote PA1,2). Postpartum, mothers explained that they did not know which activities were safe to do in the (early) postpartum period (quote PA3).

Habits—during pregnancy (PA)

Both pregnant women and fathers-to-be said that stopping PA because of sickness/discomfort (of the partner) at the beginning of the pregnancy was difficult to reverse again later on. Some participants explicitly mentioned that they did not want to make any efforts to resume the old habit (quote PA4).

Health consciousness—during pregnancy (PA)

Health consciousness was only mentioned by pregnant women, and quotes could be divided into three subcategories. Firstly, pregnant women recognized the advantage of being fit for the delivery, of which quotes were summarized under the ‘fit for delivery motivation’ subcategory (quote PA5). Secondly, pregnant women indicated changes in their PA attitude due to ‘healthiness’ of PA and the motivation to stay healthy. Thirdly, ‘wellbeing’ was quoted, referring to feeling good when participating in PA (quote PA6).

Mood and emotions—during pregnancy (PA)

Both pregnant women and fathers-to-be mentioned the influence of mood and emotions on changes in their PA behavior. Women seemed to experience worries about their body image or increasing body weight related to their pregnancy. Fathers-to-be on the other hand attributed changes in PA behavior to the stress they experienced for what would come, which took a lot of mental energy. Changes due to mental tiredness were highlighted by both pregnant women and fathers-to-be (quotes PA7,8).

Worries and safety concerns—during pregnancy and postpartum (PA)

Many pregnant women indicated feelings of uncertainty about the safety of PA. They experienced worries related to the baby and as a result did not want to take any risks. They considered some activities such as bicycling or ball games too dangerous to continue practicing (quote PA9). They were afraid of falling or to jump as the “baby might fall out”, explained that the activity might not be good for the baby or worried because their fitness level was not what it used to be (quote PA10). Postpartum, mothers stated to be afraid to restart PA due to the slow recovery process of their pelvic floor (quote PA11).

Self-licensing—during pregnancy (PA/SB) and postpartum (PA)

Some pregnant women used their pregnancy as a justification not to be physically active, which they explained as “pulling their pregnancy card”. Pregnancy made it easier to say they were tired or did not feel like being physically active and excuses such as “today is not a good day” were used. They perceived as if they could not do everything perfectly and therefore felt that it was alright not to do anything at all. This excuse was not only used in terms of PA behavior but likewise for SB. They felt they could make more allowances for themselves and should not get up when it was not necessary. Moreover, they felt people could not blame them because they were pregnant (quote PA12). Partners often approved these resting days; this gave them time for “catching up rest” from previous busy years. Postpartum, both mothers and fathers mentioned that it was easy not to do sports, as they felt they had to stay with the baby.

Self-regulation—during pregnancy and postpartum (PA/SB)

Self-regulation was divided into the following subcategories: “self-control” (during pregnancy and postpartum—PA), “self-efficacy” (during pregnancy—PA), “self-regulation” (during pregnancy—PA and SB; postpartum—PA), “weight control” (during pregnancy and postpartum—PA), “planning” (during pregnancy—SB; postpartum—PA and SB) and “habituality of PA” (postpartum—PA). Pregnant women felt difficulties in terms of self-control (quote PA13); they stated feeling less desire to be physically active. Postpartum, mothers and fathers indicated that staying at home was just the easier option. In terms of self-efficacy, pregnant women explained how difficult it was to start moving again after a period with less PA (quote PA14). During pregnancy and postpartum, both parents (-to-be) tried to encourage themselves to keep moving, which seemed difficult, especially when they had to be physically active alone. Postpartum, mothers indicated that they repeatedly lowered their PA goals. For fathers (-to-be), the (upcoming) birth of their child was a motivation to stay healthy and avoid increases in their body weight (quote PA15). For women, controlling their gestational weight gain or limiting their postpartum weight retention was a motivation to be physically active (quote PA16). Postpartum, both parents experienced difficulties with planning and fitting PA in their daily schedules and with the new rhythm they had to find with the child (quotes PA17). This was also linked to their SB; both mothers and fathers explained that if they did not plan their activities, it was just easier to stay sitting on the sofa (quote SB1). Habituality of PA was mentioned by both parents, describing that it was difficult to restart the habit of regular PA, but that once they did they felt better which in turn motivated them (quote PA18).

Parenthood perceptions and responsibilities—postpartum (PA)

Many parents experienced barriers as a result of their perceptions about parenthood and responsibilities, which prevented them from being active. These could be subdivided into “barriers to leave the child”, “barriers to ask for help”, “barriers to self-care” and “worries and uncertainties about the care for the baby”. Mothers and fathers had difficulties with their responsibilities and self-care and on how to cope with the balance of finding time to engage in leisure-time PA versus spending time with the baby. Fathers described feelings of guilt when going for leisure-time PA because this reduced the time available to spend with their child or to support their partner, which they perceived as more important (quote PA19). Both parents described feeling selfish and guilty when they had to leave the child with someone else if they wanted to take time and do something for themselves (quote PA20). Parents did not want to bother other people with babysitting when it was for leisure-time PA, as this did not feel like a valid reason for them to ask for help. Moreover, they sometimes felt they had to justify themselves when asking for someone else to take care of the baby (quote PA21). In general, parents said that it was easier for fathers to engage in leisure-time PA than for mothers. This was mostly linked to the (perceived) responsibility of the mother acting as the primary care giver (quote PA22). Mothers felt worried about the care of the baby if they would go for leisure-time PA, while fathers were uncertain about being alone with their child. They said they were not used to being alone with the baby and were consequently afraid of not doing everything right.

##### Situational (PA/SB)

Time constraints/opportunities and convenience—during pregnancy (PA) and postpartum (PA/SB)

Both women and men experienced time constraints during pregnancy. Pregnant women explained that they were busy with work or preparations before the arrival of the baby, and as a result they had the feeling that they had less time available for PA. Fathers-to-be experienced difficulties in combining PA with their working hours in combination with other personal activities before the baby would be born. Fathers-to-be, on the other hand, also found more opportunities to engage in leisure-time PA during the pregnancy of their pregnant partner. They made use of the time their pregnant partner was tired or wanted to rest (quote PA23). Moreover, they tried fitting in as much PA as possible, as afterwards when the baby would be born their time would be limited. Indeed, in the postpartum focus group discussions both parents explained that time to be physically active was very limited, or when the baby-related tasks were complete, it was too late (quote PA24). On the other hand, there were many time opportunities to walk (e.g., as a form of active transportation) with the child while being at home during parental leave.

Practical and situational constraints—during pregnancy and postpartum (PA)

Not having or finding well-fitting sports clothing (e.g., trousers or a swimsuit) was a barrier for PA during pregnancy (quote PA25). Other mentioned constraints were the wish to keep silent about the pregnancy during the first trimester or social isolation due to sickness. Stopping working early during pregnancy was a reason for less PA, as women mentioned being active at work (e.g., a standing job). Fathers-to-be did not think it was worth to continue engaging in leisure-time PA as it would be impossible to keep up anyways after the baby was born (quote PA26). Postpartum, some practical issues were mentioned such as a stressful feeling for some parents to engage in PA after commuting to work, or just practically being too late to subscribe for a course (quote PA27). First-time parents also explained encountering many organizational issues and the need for organizational adjustments within their couple. They explained that they needed to take turns to be physically active or they had to organize themselves in advance (e.g., by finding a babysitter) if they wanted to be physically active together. They needed to think about the best timing to engage in PA, choose activities without subscription (e.g., running instead of gym classes) or between different physical activities they used to do (quote PA28).

Other priorities—during pregnancy and postpartum (PA)

Both pregnant women and fathers-to-be indicated they had other priorities. They argued that because all the “to do’s” they had before the baby would arrive, doing sports was the least important one. Being there as a father once the child would be born was considered more important than their own body, and thus, doing sports (quote PA29). Additionally, postpartum parents explained about changing priorities, and as they already had many things to do, spending time on leisure-time PA was often no longer one of them (quote PA30).

Change in activities—during pregnancy and postpartum (SB)

Pregnant women explained that they were more sedentary as they had fewer activities planned. They stayed at home more as it became difficult to go outside (e.g., due to sickness in the first trimester) or because they could not go out to have (alcoholic) drinks anymore. The preparation of the child’s room, on the other hand, made expecting parents less sedentary (quote SB3). Once the child was born, the change in SB was twofold. On the one hand, more household activities meant an increase in walking around and being busy. Caring for the baby felt like an extra job, which subsequently decreased parents’ SB (quote SB4). Mothers and fathers, on the other hand, also explained that they spent more time at home and had fewer out-of-home activities to do, which increased their SB.

Work situational—during pregnancy and postpartum (SB)

Pregnant women with an active job (e.g., nurses) explained the need to sit down more often in between during the day. They sometimes received adjusted (less active) work because of their pregnancy, which made them more sedentary. Both pregnant and postpartum women reported being more sedentary once they stopped working (quote SB5).

##### Biological (PA/SB)

Physiology and physical health—during pregnancy and postpartum (PA/SB)

For many pregnant women “fatigue”, “discomfort”, “physiological changes” and “perceived physical restrictions” were barriers for maintaining their PA levels. Fatigue was a result of the energy their pregnancy took, less/poorer sleep or being already too active at work. Discomfort included dizziness, nausea, sickness in the first trimester but also discomfort as a consequence of the growing belly later in pregnancy (e.g., back pain while sitting or Braxton Hicks contractions after doing sports). Fatigue and discomfort were often mentioned determinants for pregnant women which explained a decrease or halt in their activities whilst they increased their SB (quote SB6). Even though pregnant women were said to be more sedentary because of the fatigue and discomfort they experienced, they also indicated taking frequent breaks through long periods of SB towards the end of their pregnancy. Sitting still too long felt very uncomfortable or they had to go to the toilet more often. Postpartum, in addition to mothers, fathers also experienced changes in their PA and SB behavior due to fatigue and a lack of energy (quote PA31). Backpain as a result of carrying the baby made fathers change their work position from sitting to standing. Women also experienced physiological changes, both during pregnancy and postpartum. Pregnant women adapted their PA due to an increase in heartrate, changed fitness level, longer recovery period after activity, pelvic pain, and so on (quote PA32). Additionally, general movements and certain activities (e.g., jumping, running, Pilates) became more difficult as a consequence of the extra body weight and growing belly (quote PA33). Postpartum, the recovery after the pregnancy and delivery was a barrier to moving or changing their activities (quote PA34). Some mothers stated they wanted to be physically active but were not allowed yet in the early postpartum period or experienced difficulties which kept them from being physically active (e.g., backpain, pelvic floor issues, recovery after cesarean section).

#### 3.2.2. Interpersonal Level (PA/SB)

##### Social Influence—During Pregnancy (PA/SB) and Postpartum (PA)

During pregnancy women were confronted with imposed physical restrictions (e.g., some activities were not recommended anymore), social discouragement and advice from their environment about what they should (not) do (quote PA35,36). On the other hand, they also experienced professional and social support and motivation (quote PA37). Men also described the (for some positive and for some negative) influence of social support of their pregnant partner on their own behavior. Men indicated that they were less physically active when their partner stopped being active, to show solidarity, or because they were worried and wanted to stay with their partner when she was sick (quote PA38). On the other hand, the pregnancy positively influenced sitting behavior, sitting duration, interruptions and movement of the men. The men often did more (e.g., household tasks, taking a glass of water for their partner) when the woman was sick or tired. Men sat down more often with their partner on the sofa (quote SB7). The support of their partner made sure women could rest more, which increased their SB. Additionally, external support (e.g., cleaning personnel) or professional support (e.g., medical advice to rest) also explained an increase in their SB. Postpartum, both parents experienced different (both negative and positive) influences of each other on changes in their PA behavior (quote PA39). Social activities (e.g., walking as a social activity with other young parent friends) had a positive effect on both parents. Professional advice about what was possible and allowed in the early postpartum period was described as a facilitator by first-time mothers.

##### Influence of Baby—Postpartum (PA/SB)

The baby, and as a result the new living environment of first-time parents, had an important influence on young parents’ PA and SB, both in a positive and negative way. Taking care of the baby and keeping him occupied and entertained was a reason for many parents to walk around more, and consequently to have less time for sedentary activities such as gaming or television watching. Thanks to the baby, parents also explained an increase in other activities, such as walking. For fathers, being a role model was a motivator to engage in PA (quote PA40). Parents indicated to have changed or decreased some activities due to practical constraints or impossibilities of taking the baby with them (quote PA41). They experienced many practical and organizational difficulties when they wanted to participate in sports, e.g., finding a babysitter, adjusting their work schedule and making arrangements with their partner. These practical organization issues (e.g., picking up the baby from daycare on time) made parents skip active lunch breaks or take the car instead of active transport options in order to gain time. This is in line with the adaptations to the rhythm of the baby parents felt they needed to make, which made it more difficult to find suitable moments for PA. This often resulted in staying at home more and consequently being more sedentary. Finally, many parents indicated that being with their child became the priority over PA. They often felt they had to divide time and energy between being at home with their child or elsewhere. Parents explained not being motivated to be physically active if this meant they could not spend enough time with their child. Finally, feeding the child was an important determinant which explained an increase in SB in the early postpartum period (quote SB8).

#### 3.2.3. Environmental Level (PA)

##### Meso/Macro—During Pregnancy (PA)

Pregnant women indicated that there were not enough pregnancy-proof opportunities and alternatives for doing sports. If there were, they sometimes had to wait for courses to start, courses were not close by or were at inconvenient or limited times (quote PA42).

##### Product Price—During Pregnancy and Postpartum (PA)

For many women, the high price of pregnancy-specific sport courses was a barrier to participating (quote PA43). Postpartum, costs of equipment to take the baby with you (e.g., stroller for running) was a barrier. People also indicated that they did not want to spend money on a babysitter when it was for participating in sports (quote PA44).

#### 3.2.4. Policy Level (PA/SB)

##### Government—During Pregnancy and Postpartum (PA/SB)

Pregnant women who had to stop working during their pregnancy because of exposure to certain risks in their job (Belgian policy regulation forces pregnant women to stop working when practicing certain jobs such as childcare workers) indicated this impacted their PA and SB levels as they were at home much more and thus were less active and more sedentary (quote SB9). Postpartum, Belgian regulations about parental leave entitle women to stay at home for 15 weeks, which was described as a reason for changes in PA, especially for women who have a physically active job (quote PA45).

## 4. Discussion

The aim of this study was to identify determinants of changes in PA and SB across the transition to parenthood for both first-time mothers and fathers. This was performed in parallel with the existing frameworks and classifications of determinants of changes in eating behavior during the transition to parenthood examined in the same study population [33]. Determinants were defined at four different levels, namely the individual, interpersonal, environmental and policy level.

We found strong interactions across the (sub)levels but also between the determinants of behavior, which was similar to links across levels and sublevels of the determinants of changes in eating behavior [33]. Barriers related to the individual level were most commonly cited and accounted for more than half of the determinants explaining changes in PA and SB during this transitional phase of becoming a parent. This is in line with other research on barriers of PA during pregnancy [12,28]. Overall, most individual determinants acted as barriers (e.g., “barriers to self-care”) to engage in PA or limit SB, whereas few determinants acted as facilitators (e.g., “fit for delivery motivation”).

While many biological determinants, such as fatigue or discomfort, have similarly been described as important by others [28], they have limited modifiability and can therefore not be targeted when developing intervention strategies. However, many of the maternal biological determinants can be linked to the interpersonal level. As a consequence of pregnancy-related biological limitations, pregnant women tend to stay at home more, which has in general a negative influence on their PA and SB. This, in turn, influences the fathers-to-be as well, as they seem to be sensitive to copying the pregnant partner’s behavior (e.g., resting more) which thus consequently negatively influences their own PA and SB levels. Some fathers-to-be, on the other hand, seem to interrupt their sitting behavior more often when they want to support their partner who is experiencing discomfort, fatigue or physical restrictions because of the growing belly. Another barrier mentioned by women in our study was that they did not feel like they could reach their previous fitness levels. This inhibited them from engaging in any kind of PA. Even though some women indicated being eager to start being physically active again postpartum, they encountered biological barriers such as sufficient time needed to physically recover and to feel comfortable and safe to engage in PA again, which was also shown by others [35]. Healthcare providers should therefore be trained to give tailored, personalized advice on frequency, type, intensity and duration of PA and on how to limit SB, to help women and men towards realistic, feasible PA/SB goals [36].

Consistent information and advice of healthcare providers on PA during pregnancy is important, as healthcare providers have been identified as a source of support with considerable impact [37,38]. However, we found that women do not always consider healthcare professionals’ advice, e.g., when perceived feelings of unsafety are stronger than the advice that PA is still possible. Additionally, during the postpartum period, even though many women reported they would want to restart PA, advice from healthcare professionals inhibited them when not yet physically recovered. It is true that high-impact PA (e.g., running, jumping) is not recommended in the early postpartum period; yet healthcare professionals could encourage women to engage in low-impact PA (e.g., walking, swimming). Other research has described a lack of guidance and information from healthcare professionals as a barrier for a healthy lifestyle in a safe and effective way during the postpartum period [35]. While a trusted relationship with the healthcare provider and reliable, consistent advice are important, research has shown that healthcare providers often lack the knowledge, time and skills to give appropriate advice on lifestyle behaviors during pregnancy and postpartum [36]. This often results in limited or conflicted advice given [39]. Reliable advice is, however, crucial as it is linked to many determinants at the psychological sublevel such as “PA knowledge”, “health consciousness” and “worries and safety concerns”. Future interventions should therefore target healthcare providers and educate them about PA and SB during pregnancy and early postpartum, how they can give consistent, evidence-based advice, and support and build up a trusting relationship with expecting couples.

Besides professional influence, the partner can be an important source of information, motivation and support [28]. In addition, the broader social environment of the couple may have a strong influence on expecting couples. Similar to other studies, the present study showed that lack of knowledge about PA during pregnancy of the partner, friends and family might be an important source of demotivation of PA behavior of pregnant women, as they feel judged and discouraged when still engaging in PA [38,40]. Limiting social discouragement (e.g., through a mass media campaign aimed at the general public) could be an important aspect in order to motivate pregnant women to continue participating in PA [28]. The social environment is also linked to changes in SB, as pregnant women are advised to take sufficient rest. This might again enforce psychological determinants, such as self-licensing and worries and safety concerns, which the women already experience. The support of the partner and a broader social network in the postpartum period, e.g., to look after the baby, has been described as crucial to balance newborn care with PA goals [35]. However, to tackle self-care barriers such as guilt, hesitancy to ask for help or the inability/insecurity to leave the child with their partner/social network, social support alone is not enough to encourage parents to engage in more PA. Future interventions should thus focus on barriers related to parenthood perceptions and responsibilities. Barriers to self-care and to asking for help should be reversed to self-confidence and the insight that PA is not selfish or of minor importance. Healthcare providers and the broader social environment could play an important role in supporting parents with the new demands of parenthood and help both mothers and fathers overcome the barriers related to starting up PA after the delivery.

Even though women and men already indicated being more active in household or by walking around with the baby, which was inversely linked to their SB, leisure time PA and intensity of PA were reported to decrease in the postpartum period. Situational difficulties such as time constraints were already mentioned as a barrier to being physically active by adults in general [41]. The additional aspect of the child and the new family demands will only increase issues related to time constraints. Even though women might be motivated to regain a healthy lifestyle, a lot of time is taken up by childcare; research showed that new mothers reported requiring unexpectedly large amounts of time to care for their children in addition to their normal day-to-day duties and activities [35]. This might be linked to self-management difficulties (expecting) parents seem to experience. It is thus of importance to teach parents to cope with these barriers, e.g., through improving self-regulation skills such as planning and time management.

The policy level has not previously been described as a barrier specifically linked to the pregnancy period [28], but did come forward in our study. The policy level was linked to both pregnancy and maternity leave. Belgian policy regulations force pregnant women to stop working in certain jobs such as childcare. Women are entitled to 15 weeks of pregnancy leave (of which one has to be taken one week before the due date), whereas fathers receive only 15 days (which recently changed; at the moment of data collection, this was still 10 days). This forced distribution of parental leave in the early postpartum period pushes women into a stereotypical motherhood pattern, where they become the key care-providers for their child in the early weeks after birth. This is strongly linked to the psychological determinant “parenthood perceptions and responsibilities”. Women have more time to get used to taking care of the child and feel responsible for engaging in all the care of the child, while many fathers do not feel secure enough to be alone with the child. The latter was also confirmed by mothers not feeling confident enough to leave their partners alone with the child to do sports. This might bring parents into a vicious circle where it is very hard for women to engage in PA without experiencing feelings of worries or guilt. One suggestion would be to increase paternal leave, or to give parents the opportunity to share parental leave, in analogy with the Scandinavian system [42]. Even though recommendations about PA during pregnancy and in the early postpartum period exist in Belgium [43], it is not clear whether they are actively implemented or promoted by healthcare providers in standard pre- and postnatal care. Tailored advice, not only focusing on what kind of activities can be performed (including frequency, duration, and intensity) but likewise on barriers and enablers of engaging in PA and reducing SB, should be further developed and implemented as part of pre- and postnatal care. Moreover, including fathers (-to-be) in pre- and postnatal care is recommended.

### Strengths and Limitations

The first strength of the present study is the inclusion of participants between 12 weeks of gestation and one year postpartum. This gave us the opportunity to assess determinants at more than one time point during pregnancy and postpartum, providing insights into the transition to parenthood as a whole. Second, by using two sets of focus groups, carried out during the particular periods under investigation, recall bias of participants was reduced to a minimum. The qualitative approach using focus-group discussions is another strength of this study. Focus groups are a naturalistic (i.e., close to everyday conversation) approach and include dynamic group interactions, which provides in-depth insights and encourages participants to explore and clarify perspectives. Fourth, two researchers with extensive experience in qualitative research and PA/SB supervised the data analysis. Fifth, both mixed and same-sex focus groups were used to enable a variety of opinions and interactions during the discussions.

A limitation of this study is the homogenous sample of Caucasian, mostly higher educated, heterosexual couples and perceived physically healthy participants. The inclusion of mainly health-conscious participants is a common limitation in studies investigating health behavior [44]. Although the purpose of this study was explorative in nature, we might have missed determinants due to selection bias. Neither did we specifically question whether participating parents were single or not. The focus and importance might have been slightly different for specific (e.g., single parents, same-sex parents) or vulnerable populations (e.g., teenage parents, parents with low socioeconomic status, parents without the Belgian nationality or a migrant background) or families with more children. This may limit the generalizability of our findings. Second, our qualitative research design does not allow for statistical inferences, and thus it can be a matter of chance that some determinants were quoted by women or men only. The sex-specific information added to our findings should thus be interpreted with great caution, and not interpreted per se as determinants exclusively applicable for women and/or men. Future research should further investigate the importance of these determinants in different (sub)populations of expecting and first-time parents, taking into account the difference in importance of each determinant between women and men.

## 5. Conclusions

Changes in PA and SB during the transition to parenthood are complex and can be influenced by various determinants at the same time. Individual determinants were recognized as the main barriers. While determinants at the biological sublevel are difficult to modify, determinants at the situational and psychological sublevel are more modifiable and may be targeted in interventions. A focus should go towards improving self-management skills and teaching young parents to cope with the new demands and concerns related to parenthood perceptions and barriers. First-time parents should gain insights into the importance of self-care, and need confidence that spending time on PA is not selfish. The social environment, and more specifically the healthcare providers, have an important role in guiding (expecting) parents towards healthy PA and SB levels. As men also experience significant changes in their PA and SB levels, and as fathers can play an important role in supporting their partner, focus should go towards family-based approaches targeting the couple as a whole. Moreover, changes on a policy level are needed in order to reduce maternal perceptions and barriers linked with perceived responsibility for all care of the child in the early postpartum period. In order to support both parents, implementation and promotion of PA/SB recommendations by decision makers and healthcare providers for both mothers and fathers in standard pre- and postnatal care is recommended. Future research should further investigate the importance of these determinants in different (sub)populations of expecting and first-time parents, taking the differences in importance of each determinant between women and men into account.

## Figures and Tables

**Figure 1 ijerph-19-02421-f001:**
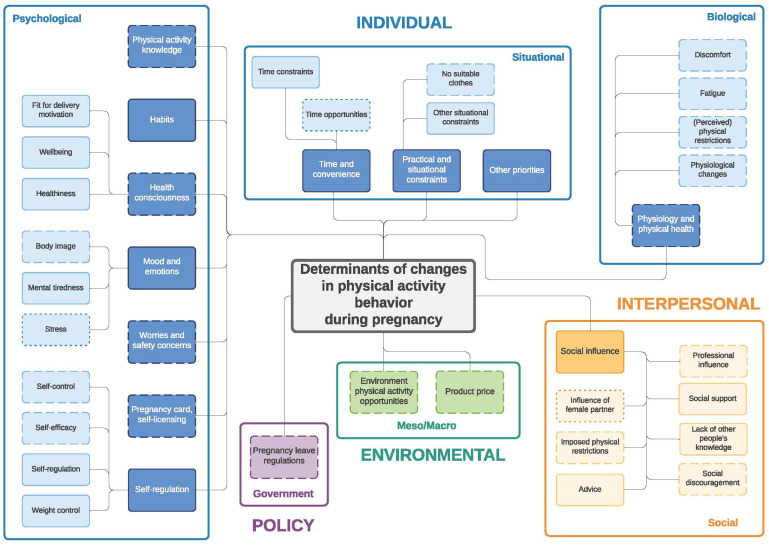
Determinants of changes in physical activity during pregnancy. Full line: determinants mentioned by both women and men; dashed line: determinants mentioned only by women; dotted line: determinants mentioned only by men.

**Figure 2 ijerph-19-02421-f002:**
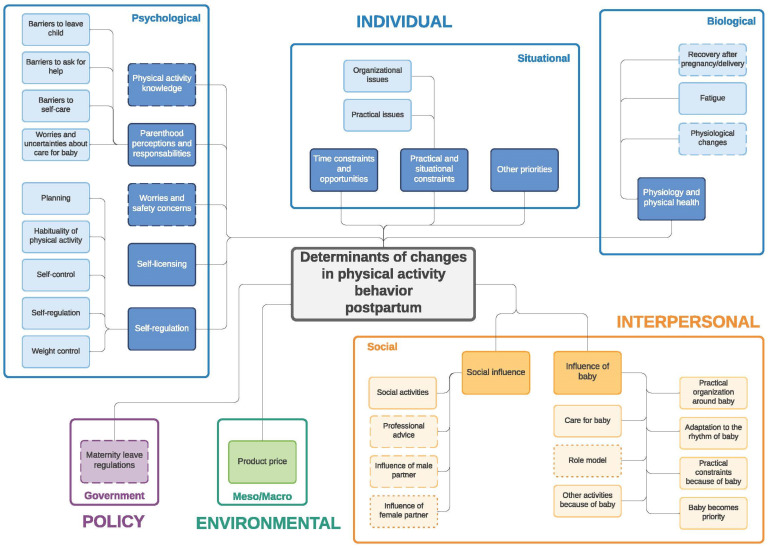
Determinants of changes in physical activity during the postpartum period. Full line: determinants mentioned by both women and men; dashed line: determinants mentioned only by women; dotted line: determinants mentioned only by men.

**Figure 3 ijerph-19-02421-f003:**
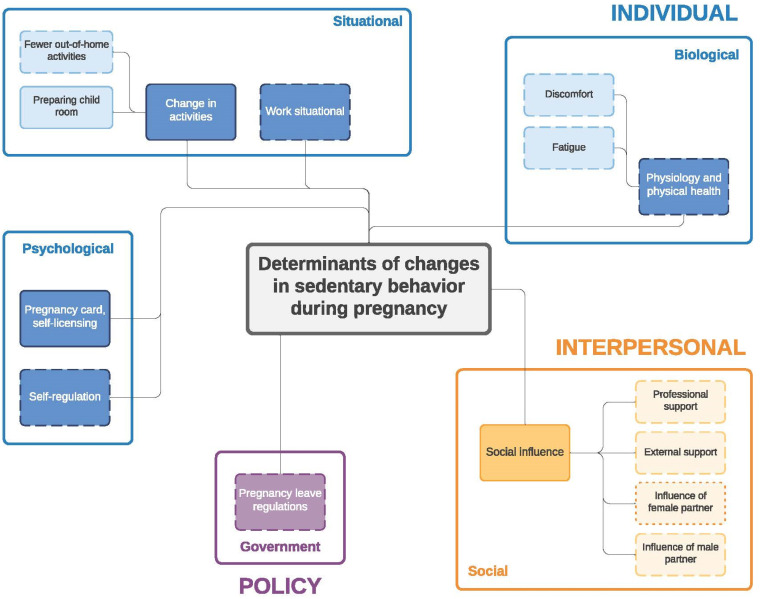
Determinants of changes in sedentary behavior during pregnancy. Full line: determinants mentioned by both women and men; dashed line: determinants mentioned only by women; dotted line: determinants mentioned only by men.

**Figure 4 ijerph-19-02421-f004:**
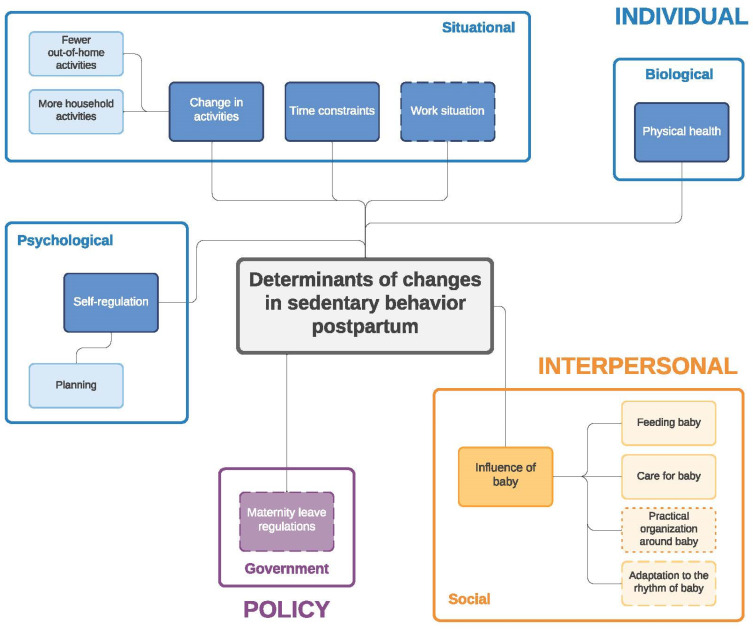
Determinants of changes in sedentary behavior during the postpartum period. Full line: determinants mentioned by both women and men; dashed line: determinants mentioned only by women; dotted line: determinants mentioned only by men.

**Table 1 ijerph-19-02421-t001:** Semi-structured question guide, including questions related to changes in physical activity and sedentary behavior.

Questions about Changes in Physical Activity and Sedentary Behavior during Pregnancy	Questions about Changes in Physical Activity and Sedentary Behavior during the Postpartum Period
1. Think about the period before pregnancy/before the pregnancy of your partner; did your physical activity/sedentary behavior change during pregnancy/during the pregnancy of your partner, and if yes, how and to what extent?	1. Think about the period before pregnancy/before the pregnancy of your partner; did your physical activity/sedentary behavior change since you became a mother/father, and if yes, how and to what extent?
2. Which factors have caused these changes in physical activity/sedentary behavior during pregnancy?	2. Which factors have caused these changes in physical activity/sedentary behavior since your child was born?
3. Which factors do you think have had the greatest influence on these changes?	3. Which factors do you think have had the greatest influence on these changes?

**Table 2 ijerph-19-02421-t002:** Participant characteristics (mean ± SD, %).

	Focus Groups on Changes in Physical Activity and Sedentary Behavior during Pregnancy		Focus Groups on Changes in Physical Activity and Sedentary Behavior during the Postpartum Period	
	Women	Men	Women	Men
Total sample (n (%))	22 (55)	20 (45)	16 (50)	16 (50)
Ethnicity (% Caucasian)	100	100	100	100
Age (years, mean ± SD)	30.1 ± 2.5	31.6 ± 2.5	30.3 ± 2.0	31.7 ± 3.5
Self-reported pre-pregnancy BMI (kg/m^2^, mean ± SD)	22.7 ± 3.1	24.0 ± 4.5	23.3 ± 4.7	25.0 ± 2.4
Respondents with a higher education (%)	81.8	75.0	93.8	87.5
Respondents reporting to be in good to very good health (%)	100	100	62.6	81.3
Respondents reporting healthy to totally healthy eating pattern (%)	77.3	80.0	93.8	62.6
Respondents reporting being active for at least 30 min/day for 5 days or more during the last 7 days (%)	49.9	45.0	6.3	37.5
% non-smokers (% ex-smokers)	100 (4.5)	100 (40.0)	100 (0.0)	100 (12.5)
Expecting parents (n)	15	14	/	/
Gestational age (weeks, mean ± SD)	28.4 ± 8.1	28.2 ± 8.6	/	/
Parents with child (n) *	7	6	16	16
Age of the newborn (weeks, mean ± SD)	9.6 ± 2.8	9.8 ± 5.2	34.8 ± 14.7	32.6 ± 15.3

* For the focus groups during pregnancy, both expecting parents as well as parents with a first child less than three months old participated. BMI: Body Mass Index.

**Table 3 ijerph-19-02421-t003:** Semi-structured question guide, including questions related to changes in physical activity and sedentary behavior.

During Pregnancy	Postpartum
**1. Decrease in PA and increase in SB**	
Decrease in PA: - Daily activity *e.g., less active because staying more at home for ‘a lazy day’* - Intensity *e.g., walking instead of running* - Leisure time PA *e.g., not going to gym classes anymore* - Transport-related PA *e.g., by car instead of bike* - Work related PA *e.g., adjusted work*	Decrease in PA: - Intensity, frequency and duration of PA - Leisure time PA *e.g., stop doing dance classes together as a couple*
Increase in SB: - Prolonged sitting time - Doing less *e.g., fewer out-of-home activities* - Extra rest *e.g., lying down during the day*	Increase in SB: - Leisure-time-related SB *e.g., resting/television watching when the baby is asleep* - Forced sitting time *e.g., when the baby falls asleep in one’s arms, while breastfeeding*
**2. Increase in PA and decrease in SB**	
Increase in PA: - Leisure time PA *e.g., more leisure-time PA*	Increase in PA: - Transport-related PA *e.g., more active transportation by bike* - Leisure time PA *e.g., doing sports when the baby is in daycare* - Household-related PA - *e.g., more walking around to do chores in the house* - Child related PA *e.g., playing with the baby*
Decrease in SB: - SB at home *e.g., preparing child room instead of sitting*	Decrease in SB: - Household-related SB *e.g., doing household tasks instead of sitting* - Leisure-time-related SB *e.g., less time to sit together in the sofa*
**3. Shift in PA and SB**	
Shift in: - Moments/timing of PA *e.g., only time for walking during the weekend* - People with whom you engage in PA *e.g., activity together with partner decreased* - Type of PA *e.g., start swimming, workouts through YouTube* - Sedentary bouts *e.g., standing up more or less often*	Shift in: - Moments/timing of PA *e.g., earlier in the morning or later at night, less spontaneous* - Organization of PA *e.g., taking turns to go for sport* - Type of PA *e.g., running instead of gym classes* - Sedentary bouts *e.g., standing up more often to change diapers or taking care of the baby*

**Table 4 ijerph-19-02421-t004:** Quotes about determinants of physical activity (PA) and sedentary behavior (SB) on the individual (psychological) level.

Subcategories of Psychological Level	Quotes during Pregnancy	Quotes during the Postpartum Period
Physical activity knowledge	Quote PA1 (pregnant woman 5): “I managed to continue exercising, because at the beginning of my pregnancy I asked a sports coach for recommendations about what I could and could not do”. Quote PA2 (pregnant women 8): “I really asked the gynecologist every time *‘I’m still jogging, is that still allowed?’* because I felt very hesitant about whether that was still allowed”.	Quote PA3 (first-time mother 16): “I am at a point that I would like to do some sports, but I don’t know what I can do”.
Habits	Quote PA4 (pregnant woman 2): “For me it was because of my pelvis. If you have to work, walk around all day, cycle home, in the evening there are cooking classes where you also have to stand up all the time; I felt this in my pelvis. Now it is better, but I don’t think I will start again doing sports”.	
Health consciousness	Quote PA5 (pregnant woman 4)—fit for delivery motivation: “They say that giving birth is top sport. I truly believe that if you can start the delivery at a good level of fitness, it makes a huge difference”. Quote PA6 (pregnant woman 22)—wellbeing: “I did these YouTube workouts. You knew that after coming home, either you stay up for 20 more minutes to move your arms and legs up and down, or you settle yourself on the sofa, not to come out anymore. And afterwards it felt good, I often felt better when I did the effort to do these exercises”.	
Mood and emotions	Quote PA7 (pregnant woman 15): “During my last trimester I did absolutely no sport activities. Because I was exhausted, both physically as well as mentally, I didn’t feel like it then”. Quote PA8 (expecting father 8): “I had to lower my exercise goals compared to what I actually wanted, because mentally you do not have the peace anymore, you do not have the mental freedom to go for it as hard as you used to do”.	
Worries and safety concerns	Quote PA9 (pregnant woman 13): “You start thinking that playing football could be pretty dangerous. If you get a ball in your belly, or you go into a duel and you get an elbow; you do not want to take such a risk”. Quote PA10 (pregnant woman 22): “In my first trimester I just couldn’t do any sports. But now, I could do e.g., Zumba again, but I don’t dare to do it yet. Because I still notice that I get tired very fast and my fitness level is not what it used to be, and two weeks ago I fainted in the supermarket. So now I am afraid, going exercising for one hour, I don’t dare to do that anymore”.	Quote PA11 (first-time mother 16): “I used to dance once a week. I stopped doing that. (…) My pelvic floor has not recovered completely yet. I have had some problems and that makes me afraid of starting again with e.g., jumping”.
Self-licensing	Quote PA12 (pregnant women 17 and 18): Pregnant woman 18: “The nice thing is also that if you cancel something because you are too tired, nobody blames you for that”. Pregnant woman 17: “Indeed, pregnancy card!”	
Self-regulation	Quote PA13 (expecting father 10 about partner)—self-control: “I go running and swimming. I try to take her with me, but I have noticed that she has less desire to come with me”. Quote PA14 (pregnant woman 10)—self-efficacy: “I stayed home and didn’t go to work because I literally was nailed to my couch or bed. Afterwards it was very difficult to start moving again. And it remains very difficult for me”. Quote PA15 (expecting father 20)—self-regulation: “This is my goal, I try to get in shape by the time the baby is there, and then I have to make sure I can maintain this”. Quote SB1 (pregnant woman 17)—planning: “Yes, we normally went to do several things, but then I was like *‘Oh, never mind, the sofa is way too attractive’*”.	Quote PA16 (first-time father 7 about partner)—weight control: “Actually, my wife increased her physical activity levels, with e.g., aqua-gym, because she realized that she gained weight. But it was more difficult to lose it again compared to earlier, back then she used to lose her weight easily when being active”. Quote PA17 (first-time father 14)—planning (PA): “I used to go running at least once or twice a week, and once swimming. Swimming is not an option anymore, running, once every second week. (…) I think that if I were to organize that, I would manage to do it again”. Quote SB2 (first-time father 11)—planning: “It is too tempting, if you are at home to just sit on the sofa and turn the television on, and turn her [the baby] around so she doesn’t face it”. Quote PA18 (first-time mother 15)—habituality of exercise: “I work in a healthcare setting, and once I started working again, my activity levels increased. (…) And I really noticed that when I restarted working, I also felt like moving more, just because I had to move again”.
Parenthood perceptions and responsibilities		Quote PA19 (first-time father 11)—barriers to leave child: “In some way I feel guilty when I’m not around enough, and it’s not because my wife tells me so, it is just for myself, when you have barely seen him [child], that is really not ok”. Quote PA20 (first-time father 4)—barriers to self-care: “If you think about it, it gives you a strange feeling, yes I want to go and do sports, but it is also very selfish, because it’s a moment that I don’t want to take care of the little one, while he [the baby] has become the focus of your life right now, because he [the baby] cannot do anything yet”. Quote PA21 (first-time mother 5)—barriers to ask for help: “Looking for a babysitter when you want to engage in sports is stupid, because it should be part of your normal life and that is nothing that requires a babysitter. So we don’t take a babysitter for that, but as a result we exercise half as much as before, so actually it should be a reason to take a babysitter, but it is a barrier”. Quote PA22 (first-time father 10 and first-time mother 16)—barriers to self-care: First-time father 10: “I think that especially women have more of a sense of responsibility, the feeling they have to be the one to take care of the child. As a man, you have more freedom and you can still do your things, your sports and hobbies. But as a woman, you have to think about all these practical things for the baby”. First-time mother 16: “I think that’s very much the case with me. I quickly feel like I’m being selfish when I do that. (…) If I want to go and do sports then I have to find a babysitter, I have to take care of his [the baby] food etc., and then I think *‘Never mind, I’ll just do it myself’*. Then I also do not have to bother someone with this. I also feel like this is my task”.

**Table 5 ijerph-19-02421-t005:** Quotes about determinants of physical activity (PA) and sedentary behavior (SB) on the individual (situational) level.

Subcategories of Situational Level	Quotes during Pregnancy	Quotes during the Postpartum Period
Time and convenience	Quote PA23 (expecting father 7 and 11)—time opportunities: Expecting father 11: “She [pregnant partner] was constantly tired, and then I just relaxed with her”. Expecting father 7: “I’ve tried not to do that, I told her ‘*You can just lie down on the sofa, and I will go and do some sports’.* So I took advantage of that and went to do something else”.	Quote PA24 (first-time father 14)—time constraints: “Before, if I came home earlier than half past 8, I would think *‘It’s still a bit early to settle on the sofa, I’m going for a run’*. And now, if I come home at half past 8, you have your child, it is nice, you are playing with him for half an hour, and then afterwards it got too late again to engage in sports activities. Playing with your child is also a nice distraction after work. I guess if I would organize myself better, I would manage to exercise again as well”.
Practical and situational constraints	Quote PA25 (pregnant woman 3)—no suitable sport clothes: “I have to invest in sports clothes. A couple of weeks ago we wanted to go to the gym, and my pants were just too tight. So I really was discouraged to go there and spend an hour working out”. Quote PA26 (expecting father 9)—other situational constraints: “Once the baby will be here, I won’t have time anyway. So I will have to choose what I’ll do, continue playing football or go running, I won’t be able to do both of these sports. So I am not going to start training for that now either, because in six months it will stop anyway”.	Quote PA27 (first-time father 1)—practical issues: “I used to go climbing. This year, I was actually too late to register. That is why I haven’t been able to do that, even though I really wanted to. I know, it is stupid. (…) It was probably because of…, I don’t want to blame [name baby], but otherwise I would have thought about it earlier”. Quote PA28 (first-time father 6)—organizational issues: “We used to dance once a week, but from a practical point of view it is not possible anymore. If you have to sacrifice one of the activities you do, it’s the dancing you choose, because then I prefer to go exercising. The dancing is nice, but it’s more like a relaxed hobby, whereas the exercising is maybe something you need more”.
Other priorities	Quote PA29 (pregnant woman 4): “All those to do’s, so yes when you go through your to-do list, yoga is the least important one on your list”.	Quote PA30 (first-time mother 5): “What I underestimated was that in advance I thought that I definitely wanted to continue exercising, because I believe doing sports is very important. But you can’t estimate that after… yes, your priorities are just completely different. You want to be with your child, you also would like to go do sports, but you just prefer to be with your child”.
Change in activities	Quote SB3 (expecting father 13): “At this moment we are busy preparing the child’s room, as we are making the cabinets ourselves, it is not so quiet anymore during the weekends, on the contrary, we’re mostly busy, especially working”.	Quote SB4 (first-time mother 4): “I will not sit down on the sofa as easily anymore as before and just watch television, because I have much more things to do in the household, and it makes me walk around much more as well”.
Work situational	Quote SB5 (pregnant woman 15): “When I was working, I was moving around all the time, and now it is already 3 weeks since I’m at home that I’m sleeping an extra hour in the morning and I also sit more throughout the day”.	

**Table 6 ijerph-19-02421-t006:** Quotes about determinants of physical activity (PA) and sedentary behavior (SB) on the individual (biological) level.

Subcategories of Biological Level	Quotes during Pregnancy	Quotes during the Postpartum Period
Physiology	Quote SB6 (pregnant woman 5)—discomfort: “At my job I don’t sit much and as a self-employed person I kept on working. But I would sit down more often in between, if I noticed that it had been a bit too much, I just sat down and let my patients do more by themselves during a therapy session. But in the evening, I had to interrupt the sitting time. If I sat down too long, I would get really restless”. Quote PA32 (pregnant woman 19)—physiological changes: “I’m already feeling shortness of breath when I go up the stairs. I’m ashamed, this is not me, where did my physical fitness level go?” Quote PA33 (pregnant woman 18)—perceived physical restrictions: “My motion space; I did not expect that my belly would have such an impact on my normal movements. Sometimes I forget that it’s there, and I crash into something or I don’t fit through somewhere I would normally fit through…”	Quote PA31 (first-time father 11)—fatigue: “Yes, fatigue is an important factor of course, going for a run with a tired body, all my trainings are difficult nowadays, that used to go better, but now my running tempo is much lower than what it used to be”. Quote PA34 (first-time mother 15)—Recovery after pregnancy: “I really do want to go back to exercising but I’m not allowed to do anything at all, I’m only allowed to swim and walk and that’s what I do”.

**Table 7 ijerph-19-02421-t007:** Quotes about determinants of physical activity (PA) and sedentary behavior (SB) on the interpersonal level.

Categories of Interpersonal Level	Quotes during Pregnancy	Quotes during the Postpartum Period
Social influence	Quote PA35 (pregnant woman 11)—imposed physical restrictions: “I have to repeat to myself *’No, you can’t go for a run any longer*‘. Even though I would still be able to do it, it is not recommended anymore”. Quote PA36 (pregnant woman 4): Social discouragement: “Those comments made me stop riding my race bike earlier than I had planned. My race bike was standing in the hallway of our office, and my colleagues were saying to each other, *’Is she still cycling, that is not acceptable.’* So that may also be a factor, for me that was the reason that made me stop doing it”. Quote PA37 (pregnant woman 1)—professional support: “I try to keep moving as much as possible. Also, my gynecologist is in favor of this. She said that if I keep moving, it might help me to have a smooth delivery”. Quote PA38 (expecting father 8)—influence of wife: “At the beginning, especially when she felt nauseous, I was worried to leave her alone and had difficulties to say ‘*ok so now I will go exercise’*”. Quote SB7 (expecting father 11)—partner support and influence of wife: “I did much more in the household at the end of her pregnancy (…) She had pain when cleaning the dishes, so I told her not to do it anymore. I had to stand up much more, to take this or that. But at the beginning of her pregnancy, we were lying down often on the sofa much more. Because she was constantly tired, and then I just went and laid down with her”.	Quote PA39 (first-time mother 13): “I would very much like to do something, but my husband is self-employed, so he gets home quite late. Then he also has his sport activities planned, soccer, tennis,... This means that I don’t have much time to do anything myself”.
Influence of baby		Quote PA40 (first-time father 16)—role model: “You feel like you have to be a role model”. Quote PA41 (first-time father 2)—practical constraints: “It is a lot of organization, for example the climbing club, you have to keep him [the baby] busy once you are there, so it is because of laziness that you stay at home and just do something else. And not to have all the hassle that comes with it”. Quote SB8 (first time mother 3)—feeding child: “At the beginning you had to sit still to feed the baby, and that means sitting still for a long time”.

**Table 8 ijerph-19-02421-t008:** Quotes about determinants of physical activity (PA) and sedentary behavior (SB) on the environmental level.

Categories and Subcategories of Environmental Level	Quotes during Pregnancy	Quotes during the Postpartum Period
Meso/Macro		
Environment constraints	Quote PA42 (pregnant woman 10): “There are not many options for pregnancy swimming. Here in [name city] you can go to the [name swimming pool], but that is only on Sundays, and that is not a good time for me”.	
Meso/Macro		
Product price	Quote PA43 (pregnant woman 1): “I searched for pregnancy yoga, but it was 14€ per lesson, it was mainly the price of this course that prevented me from doing it”.	Quote PA44 (first-time mother 5): “I want to engage in sports activities, but I do not want to spend money on a babysitter for that”.

**Table 9 ijerph-19-02421-t009:** Quotes about determinants of physical activity (PA) and sedentary behavior (SB) on the policy level.

Categories and Subcategories of Policy Level	Quotes during Pregnancy	Quotes during the Postpartum Period
Governmental regulations		
Maternity leave regulations	Quote SB9 (pregnant woman 13): “I have been home since December, I had to quit my job immediately as I am a childcare worker. It is a job during which I stand a lot, I am very busy during the day. But now, if I don’t feel like doing some household work, I sit down on the sofa and watch a film or a series”.	Quote PA45 (first-time mother 14): “I work in a healthcare setting, for me, being at work was mostly my exercise. You hardly sit down and you have to walk around a lot. Physical work as well, lifting up things. (…) I will start working again in a month, and then I think moving more will come back”.

## Data Availability

Audio tapes and transcribed interviews of the study can be retrieved through the corresponding author on reasonable request.

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
