# Peer review of "Determinants of Changes in Women’s and Men’s Physical Activity and Sedentary Behavior across the Transition to Parenthood: A Focus Group Study"

_ijerph, 2022, doi:10.3390/ijerph19042421_

Round 1

Reviewer 1 Report

This is an interesting study in which the authors make a significant contribution to research by exploring the determinants of changes in physical activity and sedentary behavior across the transition to parenthood for first-time parents.

Overall comments:

The article is methodologically sound and well presented. However, the authors need to check their sentence structure and use of prepositions. Prepositions were sometimes missing or used incorrectly making some sentences difficult to read. For example

Line 70 – 73 Sentence is ambiguous and hard to read. Suggest edit.

Line 309 Can the author clarify what they mean by ’First-time parents also explained to encounter many organizational issues ….’?

Author Response

Dear reviewers,

Thank you for your effort to review our manuscript entitled “Determinants of changes in women’s and men’s physical activity and sedentary behavior across the transition to parenthood: a focus group study”. We, the authors of this manuscript, appreciate your constructive comments which helped us to further improve our manuscript. All comments were addressed and are discussed. Furthermore, all the changes are highlighted using track changes in the originally uploaded version of the manuscript and this new version is uploaded in the submission system.

We look forward to receiving your response.

With kind regards and on behalf of the co-authors,

Vickà Versele

Reviewer 2 Report

First I would like to congratulate authors for hard work it imposses to do qualitative research. I have enjoyed reading your manuscript and I believe this piece of research is neccesary for all professionals working with pregnant and postpartum people. Please, take all the suggestions below as a humble opinion of your manuscript.

Line 45: Please, add a reference to the first sentence to support your statement.

Line 50: Please change preterm for preterm birth rates.

Line 52: Please, remove the first and; and add fetal before macrosomia.

Line 56: Please, review all mentions to physical activity to maintain the acronym PA. Also, on lines 144, 156. 175, 179, 460. As well as for sedentary behaviour: lines 183, 187, 461.

Line 66: It would be recommended to mention some of the short-term adverse health outcomes, as well as to add a reference for this statement.

Line 83: Please, rewrite last part of the sentence. Here is a suggestion: with no research available on behaviour change.

Line 92: A similar statement regarding the inclusion of heterosexual couples included in your previous paper (Versele, V., Stok, F. M., Aerenhouts, D., Deforche, B., Bogaerts, A., Devlieger, R., Clarys, P., & Deliens, T. (2021). Determinants of changes in women's and men's eating behavior across the transition to parenthood: a focus group study. The international journal of behavioral nutrition and physical activity18(1), 95. https://doi.org/10.1186/s12966-021-01137-4), is adviced.

Line 93-95: Please, review English grammar for this sentence and rewrite accordingly.

Linen 106: Please, add reference to Declaration of Helsinki 2013.

Line 110-115: Please, state whether the questions regarding perceived health and diet, and physical activity were validated questions taken from other questionnaires or scientific research, or whether they were just general questions. If possible, please add reference for these questions for replicability purposes.

Line 114: Please, delete the ‘on’ at the beginning of the question. Also how was this question quantified? Please, add this information onto the paper for replicability purposes.

Line 116: Height and mass should have been measured objectively. However, this is acceptable. Please, take this into account for future research.

Line 122: Were there any triangulation undertaken or validation of the questions included in the semi-structured interview? Please, add a justification of the use of these particular questions in your study. Regarding these questions, it is clear that the pregnant person and their partner were asked about the pregnant person’s PA and SB behaviour. However, was the pregnant person also asked for their partners PA and SB? If so, please make this clearer.

Line 146: Please, add data to highly educated and physically healthy on text.

Line 148: Please, review English grammar for table 2 title.

Line 156: Within table 3. In section 3, please rewrite: people one does PA with. It is a bit confusing.

Line 161: This section is a piece of art, amazing work done here. Just one appreciation regarding figures. Would be interesting to include whether each determinant influenced behaviour change (PA and SD) in a positive or negative manner. Especially regarding SD, for example in the social influence aspect, this could be interpreted as positive or negative, since sometimes the advice is to do less activity, hence positively influencing and promoting SB. I think this can assist the reader in their understanding of this determinants. Another example relates to change in physical activity during pregnancy/postpartum regarding the psychological section and the determinant physical activity knowledge. I would understand that this determinant positively influences physical activity during both periods, since it is known how physical activity can influence our health. However, then in the in the paper section 3.2.1.1 it is stated that lack of knowledge negatively influences physical activity. In this particular case I would suggest to change the determinant for lack of knowledge, as it reflects more accurately participants’ perceptions.

Line 163: Please, eliminate brackets to enumerate determinants. A suggestion would be: 1) determinants of changes in PA behaviour.

Line 183: Within figure 3, in the psychological section, what do the authors mean with pregnancy card? Do they mean sort of “pregnancy label”?

Line 213: Please, add a ‘the’ before stress, as without it the sentence can be misunderstood.

Line 228: Self-efficacy should be included as a determinant or sub-determinant of self-licensing, given the responses given from pregnant participants. I know this is also included in self-regulation, however it overlaps with both.

Line 243-244: This sentence is slightly confusing, since it looks it refers to postpartum women, however it mentions pregnant women. Please, change with for of.

Line 250-253: Please, review English grammar and rewrite appropriately.

Line 262: Please, change sentence for ‘the care of the baby’.

Line 274: Please, add a ‘to’ before act, or replace for ‘acting as’.

Page 12: Last quote in the postpartum period. Please, hyphen aqua-gym.

Line 302: Please, add PA after for.

Line 350: Please, separate often and mentioned.

Line 371: In general, in section 3.2.1, sometimes it is confusing to understand the explanation of the determinants, since along the paragraphs and in some sentences the authors refer to the postpartum period but suddenly mention aspects of the pregnancy. I would suggest this is written clearer, for example first explaining all determinants regarding pregnancy and then postpartum.

Line 382: Pleas, remove ‘the’ before sitting behaviour.

Line 385: Please, change settle for sitting.

Line 387: I would suggest changing the term ‘cleaning lady’ for cleaning personnel.

Line 415: Table 7, please keep same table format along the manuscript and according to publication guides.

Line 581: Please, remove behaviour.

Line 594: I know authors have vaguely referred to political adaptations and change in policies during the discussion section. However, this is mainly the first step that should be taken to reduce maternal burden in the postnatal period and to support parents (women and men) to be more physically active. I believe, this should be also mentioned in the conclusion section. I know it might be out of the scope of this paper but reducing the strategies to mainly supporting parents on how to organise themselves better is quite unrealistic and deviates to focus from what is important, which is support from the governing bodies.

Line 658: Reference 3 has no authors.

Line 677: Please, review WHO reference.

Author Response

(The authors gave the same response as above.)

Reviewer 3 Report

Interesting paper, and timely for me as I am expecting twins in the within the next week or so. I have posted some comments below, where the largest concerns are about some claims in the Intro and linking the Conclusion back to the innovative components of the paper. I also listed some orthographic suggestions. Beyond what I've listed in the orthographic section, there were just some awkward wording (especially in the Intro) so I'd recommend having another pass at this section.

Clarifications

L62: While I have no doubt that PA is important for fathers, no evidence presented to support the statement that it’s equally important for expectant mothers and fathers. I’d temper this statement.

L66: Need to cite evidence for this statement about PA not returning if it is not performed post-partum. It’s not possible that parents just wait until the babies are toddlers and then begin a PA routine? Maybe maybe not – but the strong statement here should be supported.

L70: I really like this point. Perhaps you can link to the wider literature on habit formation showing that moments of upheaval to routine are ideal for establishing new habits.  

L91: I was not familiar with the term snowball sampling and had to look it up. Could you please include a reference for others like me. Also is the word ‘purposeful’ necessary?

L225: ‘Excuse’ seems like a loaded term, especially compared to self-licensing which was used in the title. Would ‘justification’ be more neutral.

General Comments

-The paper touts same sex inclusion as a benefit, but I don’t see this mentioned anywhere in your results. Are we to infer this from influence of male/female support categories? Did you find anything specific to same sex couples?

-Similarly, in the Intro the paper indicates that the inclusion of fathers and sedentary behavior as an advance over previous studies. It’d be nice if the Discussion could specifically discuss these two features and comment on how the inclusion of these components advances the field.  

Orthography

L54: As written, this sentence Implies that this advice is for woman only, which is not true. Please reword.

L65: Awkwardly worded sentence to begin this paragraph.

L99: Comma after the word ‘discussion’

L102: Change to ‘More details describing the design…’

L119: I’d change ‘equally’ to ‘also’

L140: The statement that SPSS was used to ‘analyze’ the data is pretty generic. Here in the methods section can you tell us precisely what you did.

Tables 5-9: Seems extraneous and distracting to continually repeat the large comment about how comments were labeled. I’d do it once in Table 4 and then just briefly refer to that explanation in the other table notes.  

Line 517: ‘study’ rather than ‘studie’

Author Response

(The authors gave the same response as above.)
